

# Improved representation of phosphorus exchange on soil mineral surfaces reduces estimates of P limitation in temperate forest ecosystems

Lin Yu[1,2], Silvia Caldararu[2], Bernhard Ahrens[2], Thomas Wutzler[2], Marion Schrumpf[2,3], Julian Helfenstein[4], Chiara Pistocchi[5], and Sönke Zaehle[2]

[1]Centre for Environmental and Climate Science, Lund University, Sölvegatan 37223 62 Lund, Sweden
[2]Max Planck Institute for Biogeochemistry, Hans-Knoell-Str. 10, Jena, 07745, Germany
[3]International Max Planck Research School (IMPRS) for Global Biogeochemical Cycles, Jena, 07745, Germany
[4]Agroscope, Reckenholzstrasse 191, 8046 Zürich, Switzerland
[5]Eco&Sols, Institut Agro, CIRAD, INRA, IRD, Place Viala 34060 Montpellier cedex 2, France

*Correspondence to*: Lin Yu (lin.yu@cec.lu.se)

**Abstract.** Phosphorus (P) availability affects the response of terrestrial ecosystems to environmental and climate change (e.g. elevated $CO_2$), yet the magnitude of this effect remains uncertain. This uncertainty arises mainly from a lack of quantitative understanding of the soil biological and geochemical P cycling processes, particularly the P exchange with soil mineral surfaces, which is often described by a Langmuir sorption isotherm.

We first conducted a literature review on P sorption experiments and terrestrial biosphere models (TBMs) using Langmuir isotherm. We then developed a new algorithm to describe the inorganic P exchange between soil solution and soil matrix based on the double-surface Langmuir isotherm and extracted empirical equations to calculate the sorption capacity and Langmuir coefficient. We finally tested the conventional and new models of P sorption at five beech forest sites in Germany along a soil P stock gradient using the QUINCY (QUantifying Interactions between terrestrial Nutrient CYcles and the climate system) TBM.

We found that the conventional (single-surface) Langmuir isotherm approach in most TBMs largely differed from P sorption experiments regarding the sorption capacities and Langmuir coefficients, and simulated a too low soil P buffering capacity. Conversely, the double-surface Langmuir isotherm approach adequately reproduced the observed patterns of soil inorganic P pools. The better representation of inorganic P cycling using the double Langmuir approach also improved simulated foliar N and P concentrations, and the patterns of gross primary production and vegetation carbon across the soil P gradient. The novel model generally reduces the estimates of P limitation compared to the conventional model, particularly at the low-P site, as the model constraint of slow inorganic P exchange on plant productivity is reduced.



## 1 Introduction

Nutrient availability is one of the key factors affecting the productivity of terrestrial ecosystems and their future carbon (C) balance (Fernández-Martínez et al., 2014; Wieder et al., 2015). Although nitrogen (N) is the main constraint of plant biomass responses to elevated $CO_2$ (eCO2) concentration in many terrestrial ecosystems, phosphorus (P) availability likely constrains the biomass responses to eCO2 in major global biomes (Du et al., 2020; Elser et al., 2007; Lebauer and Treseder, 2008; Terrer et al., 2019). For instance, the tropical forests and forests grown on old soils are known to be strongly limited by P availability,

and might not be able to sequester additional C in the future as the $CO_2$ concentration continues to increase (Hubau et al., 2020; Jiang et al., 2020). While temperate and boreal forests are generally considered N limited, recent studies have shown a decreasing P nutritional status is concomitant with increasing atmospheric $CO_2$ (Penuelas et al., 2020; Jonard et al., 2015). These findings highlight the importance of representing C-P interactions in terrestrial biosphere models (TBMs) and including P cycle processes in the future global C balance predictions.

There has been a continuous effort to include P cycling processes into TBMs in the past decade (Goll et al., 2012; Goll et al., 2017; Sun et al., 2020; Thum et al., 2019; Yang et al., 2014; Zhu et al., 2019; Wang et al., 2010; Yu et al., 2018). Many current TBMs employ the scheme (Fig.1a) developed by Wang et al. (2007) to describe soil geochemical processes. This model considers the soil inorganic P (Pi) as soluble Pi ($P_{sol}$), labile Pi ($P_{lab}$), sorbed Pi ($P_{sorb}$), occluded Pi ($P_{ocl}$) and primary Pi ($P_{primary}$), with few exceptions where labile and sorbed Pi pool are grouped into one pool (Zhu et al., 2019; Zhu et al., 2016). The

exchange between soil solution and soil matrix is described with a Langmuir adsorption isotherm assuming that $P_{sol}$ quickly exchanges with $P_{lab}$ on the mineral surface. The $P_{lab}$ pool also slowly exchanges with $P_{sorb}$ at a linear rate regardless of $P_{sol}$ concentrations, but this relationship is parameterized very differently across TBMs (Helfenstein et al., 2020).

Many modelling studies emphasize the significance of biological P processes (Fleischer et al., 2019; Jonard et al., 2010; Yu et al., 2018; Wang et al., 2010), i.e. organic P recycling in terrestrial ecosystems, but the role of geochemical P processes is less

discussed and remains unclear (Sun et al., 2020). Particularly, the effect of (ad)sorption kinetics was seldom discussed in previous modelling studies (Fleischer et al., 2019; Yang et al., 2014), although they are known to directly and strongly regulate $P_{sol}$ and $P_{lab}$, and thus affect P bioavailability (Frossard et al., 2000; Shen et al., 2011).

In this paper, we first conducted an extensive literature review of important soil characteristics affecting soil phosphorus sorption kinetics, and then developed and applied a novel model concept to reconcile measured P stocks with model

simulations. In this new model, we applied the "two-surface" modification (Holford and Mattingly, 1976; Mcgechan and Lewis, 2002) to the conventional single-surface Langmuir isotherm to formulate a novel algorithm for Pi exchange with mineral surfaces, namely double-surface Langmuir isotherm. We hypothesised that both $P_{lab}$ and $P_{sorb}$ exchange with $P_{sol}$ in the new model (Fig. 1b). We then compared and evaluated the performances of the novel and conventional models with measured soil Pi pools as well as foliar N and P concentrations for a gradient of soil P stocks (164–904 g P/ $m^2$) and availability in a

similar climate (MAT 4.5–8 °C) and vegetation conditions (mature German beech forests, 120–140 yr). Lastly, we tested the





sensitivity of the alternative Pi exchange schemes on ecosystem P and C cycling to changes in P cycling parameters, and tested the responses of them to changes in P availability (P fertilisation), C availability ($CO_2$ fertilisation) as well as their combination.

## 2 Methods

### 2.1 Literature review on Langmuir isotherm

The two parameters in Langmuir isotherm, maximum sorption capacity ($S_{max}$) and coefficient ($K_m$), are statistically fitted from measured adsorption curves of batch sorption experiments (Barrow, 1978). Following the scheme of Lloyd et al. (2001), most TBMs adapted the constant-equilibrium assumption to calculate a partition coefficient, $k_p$ (Eq.2), which is the ratio of change in $P_{sol}$ concentration to the change in available P ($P_{sol}$ plus $P_{lab}$). The partition coefficient affects the rate of $P_{sol}$ released from the $P_{lab}$ pool as $P_{sol}$ is consumed, thus eventually affects the soil P supply to plants and microbes.

$$P_{lab} = S_{max} \frac{P_{sol}}{K_m + P_{sol}} \tag{1}$$

$$k_p = \frac{\frac{dP_{sol}}{dt}}{\frac{dP_{sol}}{dt} + \frac{dP_{lab}}{dt}} = \frac{(P_{sol} + K_m)^2}{(P_{sol} + K_m)^2 + S_{max} K_m} \tag{2}$$

To better understand the characteristics of phosphate exchange between solution and soil mineral surface, we conducted a literature review of batch phosphate sorption experiments that describe the P sorption curves using Langmuir isotherm, and converted the fitted Langmuir parameters ($Q_{max}$, mg P/ kg soil, and $K_L$, L/ mg P, see Supplementary material Sect. S2) to

sorption capacity and Langmuir coefficient and units commonly used in TBMs ($S_{max}$ and $K_m$, g P/ $m^2$ in Eq.1). From the batch phosphate sorption experiments found in the literature, we calculated the partition coefficient ($k_p$) following Eq.2. Similarly, we reviewed the values of $S_{max}$ and $K_m$ in modelling studies and calculated $k_p$ accordingly.

To showcase the difference of Langmuir isotherm parameter values in TBMs and experiments as well as the differences between single- and double-surface Langmuir isotherms, we compared the exchangeable soil Pi curves of different TBMs,

batch experiments data, to the double-surface Langmuir isotherm. We also simulated a desorption experiment with the single- and double-surface Langmuir isotherms to demonstrate their different responses to P removal.

### 2.2 QUINCY Model

QUINCY is a terrestrial biosphere model of coupled C, N, and P cycles as well as energy and water processes (Thum et al. 2019). The model represents the growth of vegetation and turnover of litter and soil organic matter at half-hourly timescales,

coupled with the calculation of the terrestrial energy and water budgets. Vegetation is represented by average individuals of plant functional types (here a temperate broadleaved tree, see Section 2.3). Gross carbon uptake, leaf area and vegetation biomass and structure are directly influenced by nutrient availability through their effect on photosynthesis, tissue growth, allocation and mortality. The model explicitly considers depth profiles (discretized into 15 layers with a default total soil column depth of 9.5 m) of soil temperature, moisture and biogeochemical pools, representing litter and soil organic matter



compartments, as well as inorganic forms of N and P. Vertical transport processes include diffusion and bioturbation. The phosphate exchange between soil solution and soil matrix is described using a conventional single-surface Langmuir isotherm, which is implemented the same way as in other TBMs. Different from other TBMs, QUINCY estimates $S_{max}$ and $K_m$ using soil texture and organic matter content (Thum et al. 2019).

To test our hypothesis that both $P_{lab}$ and $P_{sorb}$ exchange with $P_{sol}$, we modified the QUINCY Pi pool structure to describe the
Pi exchange between solution and soil matrix (Fig.1b), using a double-surface Langmuir isotherm (Eq.3) (Holford & Mattingly 1976, McGechan & Lewis 2002).

$$P_{lab} = S_{max,1} \frac{P_{sol}}{K_{m,1} + P_{sol}} \tag{3.1}$$

$$P_{sorb} = S_{max,2} \frac{P_{sol}}{K_{m,2} + P_{sol}} \tag{3.2}$$

$$S_{max} = S_{max,1} + S_{max,2} \tag{3.3}$$

$$k_m = \frac{\frac{S_{max}}{hl} - 2P_{sol} \pm \sqrt{\frac{S_{max}^2}{hlp^2} - 4\frac{S_{max}P_{sol}}{hlp}}}{2}, \tag{3.4}$$

where

$$hlp = \frac{S_{max,1}K_{m,1}}{(K_{m,1} + P_{lab})^2} + \frac{S_{max,2}K_{m,2}}{(K_{m,2} + P_{lab})^2} \tag{3.5}$$

The first sorption sites are responsible for the fast exchange (Eq. 3.1) between $P_{sol}$ and $P_{lab}$ and have much lower bonding strength than the second sorption sites that are responsible for the slower exchange (Eq. 3.2) between $P_{sol}$ and $P_{sorb}$. We
estimated two soil sorption maxima ($S_{max,1}$ and $S_{max,2}$, Eq. 3.3) and calculated two Langmuir coefficients ($K_{m,1}$ and $K_{m,2}$, Eqs. 3.4 and 3.5) based on batch experiments data from the literature review, assuming both single- and double-surface isotherms can be fitted with batch experiments data. Detailed derivation is described in Supplementary material Sect. S3.

**2.3 Sites and data**

We performed analysis at five mature beech forest stands in Germany, Bad Brückenau (BBR), Mitterfels (MTF), Vessertal
(VES), Conventwald (COM), and Lüss (LUE)) (Table 1, Lang et al. 2017). Total soil P stocks (g P/m², up to 1 m depth) decrease strongly along the gradient: BBR (904) > MTF (678) > VES (464) > COM (231) > LUE (164).

Soil was sampled up to 1 m depth at each site, with layer depths of 5–10 cm, for the measurements of total C, N, and organic and inorganic P and other physio-chemical properties such as soil texture, pH, and oxalate-extractable Fe and Al. Modified Hedley fractionations (Tiessen and Moir (2008)) were conducted on soils from all depths for the measurements of labile Pi (P
resin and Pi NaHCO₃), sorbed Pi (Pi NaOH), occluded Pi (P residual (acid digestion)) and primary P (P 1 M HCl). Beech leaves were sampled in July/August from five trees for the measurements of leaf N and P concentrations.





### 2.4 Model setup, experiments, and evaluation

### 2.4.1 Model setup

To compare the performances of different phosphate exchange schemes, we applied here the conventional single-surface
Langmuir isotherm (**siLang,** Fig.1a) as well as the novel double-surface Langmuir isotherm (**dbLang,** Fig. 1b) as model
variants. To further test the causes for differences between these model variants, we introduced a **Control** simulation, which
serves as a reference run without P limitation, and a **4pool** simulation (**4pool**, Fig. 1c), which serves as a special case of **siLang**
that does not include slow Pi exchange. All four models employed Eq.3 to calculate the $S_{max}$ and $K_m$ in the Langmuir isotherms.
In **siLang** and **4pool**, total sorption capacity ($S_{max}$, Eq.3.3) only refers to $P_{lab}$ (or $P_{exchangeable}$), whereas in **dbLang** the two
sorption maxima ($S_{max,1}$ and $S_{max},2$) refers to $P_{lab}$ and $P_{sorb}$, respectively. In the **Control** model, $P_{sol}$ was kept at concentrations
not limiting plant uptake or SOM decomposition. All models were initialized with a 15-layer soil column of 9.5 m which has
a decreasing SOM content as soil depth increases (Thum et al., 2019). As for the inorganic P pools, the initial values of top 1
m soil were prescribed from the Hedley fractionation measurements and extrapolated to the deeper soil assuming an increased
fraction of primary P and a decreasing fraction of exchangeable soil Pi pools ($P_{lab}$ and $P_{sorb}$). The initialization of soil inorganic
P pools deeper than 1 m is described in Supplementary material Sect. S4.

### 2.4.2 Model experiment protocol

The models were spun up for 500 years with meteorology and other atmospheric forcing (atmospheric $CO_2$, as well as N and
P deposition), which are randomly drawn from the years from 1901 to 1930. During the model spinup, the P cycle was
simulated dynamically, but the more stable Pi pools, i.e. $P_{ocl}$, and $P_{primary}$, were kept constant to ensure all the models initialized
at the same field P status. After spinup, all models were run from 1901 to 2015 using the annual values for atmospheric $CO_2$,
N (Lamarque et al., 2010; Lamarque et al., 2011) and P (Brahney et al., 2015; Chien et al., 2016) deposition, and the
meteorology (Viovy, 2018) of the respective year. We used the same maximum biological N fixation rate for all study sites
after calibration. The CN and CP ratios of the slow SOM pool were calibrated per site per soil depth to match the measured
soil CN and CP ratios so that the simulated SOM cycling adequately represented the measured site condition.

### 2.4.3 Model evaluation

Trend analyses were carried out with the Mann–Kendall test (M-K test) of the Kendall R package (Mcleod, 2011). In the
Mann–Kendall test, the tau ($\tau$) value varies between –1 and 1, where –1 represents a decreasing trend and 1 represents an
increasing trend. The modeled soil profile against the measured soil profile was evaluated with a normalized root mean square
ratio term, $K_{nrmsr}$, which is modified to represent the average proportions between modeled and measured values (Yu et al.,
2020b).

$$K_{nrmsr} = \sqrt{\frac{\sum_1^n K_i^2}{n}}, where \ K_i = min\left(\frac{Mod_i}{Meas_i}, \frac{Meas_i}{Mod_i}\right) \tag{4}$$





$K_i$ is the variable representing the ratio between simulated and measured values (in volumetric units) at the measured $i_{th}$ layer. A paired t-test was conducted between the $K_{nrmsr}$ of all study sites between **siLang** and **dbLang** to verify if one model is statistically better than the other one.

**2.5 Sensitivity analysis**

To evaluate the response of alternative Langmuir isotherms to changes in P cycling processes, we further tested the sensitivity of the **siLang** and **dbLang** models to the P cycling parameterisation at one low-P site, Conventwald [COM], and one high-P site, Mitterfels [MTF], using a standard Latin hypercube design (LHS, Saltelli et al., 2004). We selected 16 parameters that directly control flux rates of inorganic or organic P cycling processes and varied each parameter by ±20% of its default value
(Table S2) using LHS sampling of a uniform distribution, to form a set of 1000 LHS samples. The model outputs were evaluated in terms of GPP, foliar N and P contents, vegetation C stock, contents of SOC, total soil organic P (Po) and inorganic P, and ratio between $P_{lab}$ and exchangeable Pi. We measured parameter importance as the rank-transformed partial correlation coefficients (RPCCs) to account for potential non-linearities in the association between model parameters and output (Zaehle et al., 2005; Saltelli et al., 2004).

**2.6 Simulated CO2 and P fertilization experiments**

To test the effects of phosphate exchange schemes to environmental changes, we conducted a $CO_2$ fertilization model experiment, a P fertilization model experiment, and a $CO_2$-P (CP) fertilization model experiment using **siLang**, **dbLang**, and **4pool** models at each study site. In the $CO_2$ fertilization experiment, the atmospheric $CO_2$ concentration was increased by 200 ppm from 2006 to 2015. In the P fertilization experiment, 50 kg/ha $KH_2PO_4$, i.e. 1139.7 mg P/m², was added once to the soil
as soluble phosphate on Sep 16, 2006. The CP fertilization experiment is a combination of $CO_2$ and P fertilization experiments.

**3 Results**

**3.1 Langmuir isotherm parameters in batch sorption experiments and TBMs**

In batch sorption experiments, the maximum soil P sorption capacity $S_{max}$ ranges from 187 to 829 g P/m² (25-quantile and 75-quantile values, median 390 g P/m²) and the Langmuir coefficient $K_m$ ranges from 0.21 to 4.5 g P/m² (median 0.93 g P/m²),
therefore the calculated partition coefficient $k_p$ varies between 0.005 and 0.022 (median 0.01) (Table 2 and Fig. S1). Regarding the parameterization of $S_{max}$ and $K_m$ in TBMs, a few studies were directly retrieving values from sorption experiments (Wang et al., 2007; Yang et al., 2014), while most modeling studies estimate $S_{max}$ and $K_m$ based on soil types (Wang et al. 2010, Goll et al. 2012, Zhu et al. 2019) or soil texture (Thum et al. 2019).

The two studies using lab derived Langmuir parameters fitted well in the range of experimental values, but for those modeling
studies that estimate Langmuir parameters, only the QUINCY model (Thum et al. 2019) produced reasonable Langmuir


parameters and $k_p$ values; while most other TBMs greatly overestimated $K_m$ thus generating much higher $k_p$ values than batch experiments (Table 2).

It is also shown in Fig. 2a that most TBMs are at or below the lower range of curves from batch sorption experiments data (BED) except the two isotherms from QUINCY that are close to the BED-median curve. All the other TBMs either have very low $S_{max}$ or too high $K_m$, which can both lead to flattened soil Pi curves, indicating very low amounts of P are stored as exchangeable Pi in the soil even at high $P_{sol}$ concentrations.

However, the responses to P removal are noticeably different between the single- and double-surface Langmuir (**siLang** and **dbLang**) given the similar soil Pi curves (Fig. 2b). The $P_{sol}$ in **siLang** almost reached zero after a 30-day desorption experiment while the **dbLang** still maintained a $P_{sol}$ level of 0.2 g/m$^2$, suggesting a much stronger buffering capacity in **dbLang** compared to **siLang**. This is mainly because the replenishing of $P_{sol}$ in **siLang** is strongly limited by the rate of desorption from $P_{sorb}$ to $P_{lab}$, but in **dbLang** the $P_{sol}$ directly exchanges with $P_{sorb}$ when the P is removed.

**3.2 Simulated and measured ecosystem properties**

The **siLang**, **dbLang**, and **4pool** models can adequately reproduce the measured SOC content, and SOM CN and CP ratios along soil profiles at all the study sites after site- and depth-specific calibration (Figs. 3a–c, Table S1). The performance of the three models did not differ much regarding the simulation of SOM profiles (Tables 2 and S1). However, the novel **dbLang** model better reproduced the ratio between $P_{lab}$ and exchangeable Pi (*Lab-to-Exchangeable P*) than **siLang** ($K_{nrmsr}$ improvement 0.130±0.077, p < 0.05) (Table 3). The improvements of **dbLang** in modelling the labile ($K_{nrmsr}$ improvement 0.030±0.147, p > 0.05) and sorbed Pi ($K_{nrmsr}$ improvement 0.096±0.137, p > 0.05) pools were not as significant as that in modelling the *Lab-to-Exchangeable P* ratio (Table 3, Fig. 3d–f).

The simulated average foliar N content of four models (23.35±1.59, 23.05±1.84, 22.45±2.35, 22.74±2.27 mg N/g d.w. for **Control**, **dbLang**, **siLang**, and **4pool**, respectively) were within the average range of measured values (24.32±1.43 mg N/g d.w.). However, the simulated decreasing trend of foliar N content (M-K test, τ = -0.6, -0.8, -1, -0.8 for **Control**, **dbLang**, **siLang**, and **4pool**, respectively) along the soil P gradient was not found in the measured data (M-K test, τ = -0.2) (Fig. 4a). Decreasing trends in simulated foliar P content in all models (M-K test: τ = -0.8, -0.95, -1, -1 for **Control**, **dbLang**, **siLang**, and **4pool**, respectively) were instead also seen in measurements, but the decreasing trend was much weaker (M-K test, τ = -0.6) (Fig. 4b). The simulated foliar P content was highest in the **Control** model (1.40±0.09 mg P/g d.w.) and lowest in **siLang** (0.95±0.21 mg P/g d.w.) at each study site, as **Control** is not P limited and the strongest P limitation occurs in **siLang**. The simulated difference in foliar P content across models is more a reflection of differing plant P uptake than productivity (Figs. S2–5). For example, although the foliar P contents of four models at the P-rich BBR site were different (Fig. 4), the simulated gross primary productivity (GPP), leaf area index (LAI), aboveground C, and fine root C for the three models were almost identical (Fig. S2). In contrast, at the P-poor LUE site, the differences among **siLang**, **dbLang** and **4pool** in GPP, LAI, and plant C were more pronounced than that in foliar P content, because of the effect of limiting P availability on plant growth.



### 3.3 Sensitivity analysis

Our sensitivity tests at the low-P COM site (231 g P/m$^2$) and the high-P MTF site (678 g P/m$^2$) showed a diverging effect of

model choice and parameterization on selected ecosystem properties, such as GPP, plant C, and foliar P content (Figs. 5 and S6). Both models were very consistent in maintaining the vertical pattern of the simulated ratio between labile and exchangeable Pi (*Lab-to-Exchangeable P*) across different parameterizations (Fig. S7). In other words, the better fit of **dbLang** in capturing the measured decreasing trend in *Lab-to-Exchangeable P* was robust against model parameter choice (Figs. S6 and S7).

The P cycling parameters with the strongest effect on the selected ecosystem properties were very different between the two models and moderately different between the two sites (Table S3 and Fig. 5). In **siLang**, the highest impacting parameters for most selected outputs (e.g. Foliar P, GPP, and plant C as in Fig. 5) was the absorption and desorption rate between $P_{lab}$ and $P_{sorb}$ ($k_{abs}$ and $k_{des}$), i.e. the slow exchange process, while in **dbLang** they were the turnover rates and N: P ratios of slow and fast SOM pools ($\tau_{slow}$, $\tau_{fast}$, SOM_np, and microbial_np). One common feature for both models is that the low-P site, COM,

was more affected by SOM_np, maximum weathering rate (k_weath_mineral) and maximum plant P uptake rate (vmax_uptake_p_p4) compared to the high-P site, MTF, inferring to a tighter P cycle in low-P ecosystem than high-P ecosystem.

### 3.4 Modeled ecosystem responses to C, P, and CP fertilizations

The simulated ecosystem responses to $CO_2$ fertilization, P fertilization, and CP fertilization differed greatly among study sites,

models, and fertilization types due to the differences in soil P availability and model schemes of Pi exchange (Figs. 6, 7, and S3–5). At the P-rich BBR site, the simulated GPP, LAI, aboveground and fine root C, plant uptake of N and phosphate increased after $CO_2$ and CP fertilizations, but did not change after P fertilization in all three models (**siLang**, **dbLang**, and **4pool**, Fig. 6a). It indicates that the simulations at BBR were not limited by P, which is also supported by the small difference of responses between $CO_2$ and CP fertilization experiments. In contrast, at the P-poor LUE site, the simulated responses to P

and CP fertilization were much stronger than those after $CO_2$ fertilization in all three models (Fig. 6c), indicating a strong P limitation at LUE.

This simulated P stress among models can be quantified when compared to the **Control** model (Fig. 6d). The **siLang** model simulated the lowest plant P uptake at all three sites with high-, moderate- and low-P in soil. This difference in plant P uptake was only reflected in foliar P concentration at high-P site, but also reflected in other vegetation properties at moderate- to low-

P sites. The **4pool** model, which is a special case of the single-surface Langmuir isotherm, simulated similar P stress to **dbLang** at high- and moderate-P sites, while at the low-P site, simulated a high P stress as **siLang** (Fig. 6d).

The changes of model P pools after $CO_2$, P, or CP fertilizations further illustrated the effects of Pi exchange schemes on ecosystem responses under varying soil P availability (Fig. 7). In the $CO_2$ fertilization experiment, the chronic increase of $CO_2$ led to increases in plant biomass P at all study sites in all three models (Fig. 7, top panel). The concurrent increases in plant P



pools were compensated by decreases of labile and sorbed Pi as well as Po in the microbial (fast SOM) pool. At the high-P
       BBR site, the increases in plant P ($631.4\pm1.4$ mg P/m$^2$), litter P ($107.8\pm0.8$ mg P/m$^2$), slow Po in SOM ($62.8\pm0.3$ mg P/m$^2$)
       were similar in all models, and the mobilization from soil exchangeable Pi ($417.4\pm16.1$ mg P/m$^2$) and fast Po SOM ($354.2\pm2.5$
       mg P/m$^2$) contributed similarly as P sources. Conversely, at the P-poor LUE site, **dbLang** simulated a significantly higher
       plant P increase ($188.3$ mg P/m$^2$) than **siLang** ($69.5$ mg P/m$^2$) and **4pool** ($18.5$ mg P/m$^2$), due to its stronger capacity of

mineralizing P from microbial P pool. Because **dbLang** can maintain a much higher SOM pool in topsoil (Fig. 3a) as $P_{sol}$ is
       better buffered under low soil Pi (Fig. 2b) compared to **siLang**, thus more P can be mineralized under eCO2.
       In the P fertilization experiments, the fate of added P largely differed between **siLang** and **dbLang**, as the added P was
       preferably and quickly transferred to $P_{sorb}$ in **dbLang** compared to **siLang** (Fig. 7, bottom panel).
       In the CP fertilization (Fig. 7, middle panel), plant P did not gain more P at high- and moderate-P sites (BBR and VES)

compared to the C fertilization, indicating both sites might not be P limited under eCO2. Surprisingly, microbes (fast SOM)
       also didn't benefit from P addition on top of eCO2 at BBR and VES, as most added P were transferred to the soil Pi pools.
       However, at the low-P LUE site (Fig. 7, last three columns), the combination of C and P fertilization produced higher plant P
       increases ($928.5$, $1202.3$, and $971.1$ mg P/m$^2$ for **4pool**, **dbLang** and **siLang**, respectively) than the P fertilization alone in all
       three models, inferring a very strong P limitation on C sequestration at the low-P LUE site.

## 255 4 Discussion

### 4.1 Model representation of soil Pi cycling and Langmuir sorption

The majority of terrestrial biosphere models (TBMs) describe the soil inorganic P (Pi) cycling processes with a similar structure
of pools and fluxes (Figs. 1a or 1c), and mostly describe the key soil Pi exchange process, i.e. the (ad)sorption and desorption
between soil solution and matrix, using the Langmuir isotherm. In this study, we have shown that the values of Langmuir

parameters in most TBMs, namely maximum P sorption capacity ($S_{max}$, Eq.1) and half-saturation Langmuir coefficient ($K_m$,
       Eq.1), largely differ from those reported in the P sorption batch experiments (Table 2). This disagreement between TBMs and
       experiments leads to much lower exchangeable soil Pi ($P_{lab}$ plus $P_{sorb}$) contents in TBMs than experiments under the same
       soluble Pi ($P_{sol}$) (Fig. 2a). It is in line with the much higher partition coefficient ($k_p$, Eq.2) values in most TBMs than
       experiments (Table 2). Additionally, the reported $S_{max}$ have a comparable size as the total soil Pi in experiments (Fig. S1),

indicating that most of the soil-associated Pi could be exchanged with the soil solution at a certain point. This implies that the
       Langmuir isotherm can describe not only the fast exchange between $P_{sol}$ and $P_{lab}$ but also the slower exchange with $P_{sorb}$. This
       hypothesis is supported by the classical sorption model of Barrow (Barrow, 1978, 1983) and the isotopic exchange kinetics
       conceptual model (Fardeau, 1995; Frossard and Sinaj, 1997; Morel et al., 2000) that Pi ions located in the soil matrix are
       distributed along a continuum of solubility, some being in rapid equilibrium with Pi ions in soil solution and some being in

slow equilibrium.

<br>




Following the concept of the soil P solubility continuum, we have developed the double-surface Langmuir isotherm (**dbLang**, Fig. 1b and Eq. 3) in the QUINCY TBM and parameterized it with the batch sorption experiments data. The double-surface Langmuir isotherm simulated a higher P buffering capacity than single-surface Langmuir isotherm (**siLang**) when P is removed from the soil (Fig. 2b), which is in agreement with the experimental results by Roberts and Johnston (2015). By comparing

**dbLang** with the conventional **siLang** model and a special four-Pi-pool structure of **siLang** (**4pool** model, used in Zhu et al. 2016, 2019), we further investigated the effects of different sorption-desorption schemes on soil P under the QUINCY TBM framework (Fig. 3). The model performance in simulating the relationship between soil labile and sorbed Pi pools (*Lab-to-Exchangeable P*) has been greatly improved by **dbLang** (Figs. 3d–f, Table 3) at our study sites of varying soil P stocks, soil texture and stoichiometry (Lang et al., 2017; Yu et al., 2020b). The improvement of **dbLang** in reproducing the *Lab-to-*

*Exchangeable P* ratio was not simply caused by improvements in modelling the individual labile or sorbed Pi pool, but rather by the improved representation of the Pi exchange among $P_{sol}$, $P_{lab}$, and $P_{sorb}$, as such an improvement is independent of site conditions and model parameterization (Figs. 3 and S7).

In this study, we have included the Hedley soil P data as part of the model validation, which, to our knowledge, has seldom been done before, although the Hedley soil P data (Yang and Post, 2011; Yang et al., 2013) have been widely used as

initialization data for TBMs. The depth-explicit and quantitative evaluation of model Pi pools with field measurements provides us a unique tool to diagnose different sorption-desorption schemes, and warrants future applications in model-data comparison or model-intercomparison studies.

**4.2 Model performance and confidence**

We have shown that **dbLang** performed better than **siLang** (and **4pool**) in reproducing the measured soil Pi pools and foliar

P contents (Figs. 3 and 4, Table 3). Several clues point toward the novel **dbLang** model producing more realistic patterns in GPP and vegetation C, although there is no direct evidence. First of all, Lang et al. (2017) have found that there are no trends in the measured tree heights or stand biomass along our soil P gradient (Table 1). Since all sites are unmanaged beech stands with a similar age structure, we can assume that GPP and vegetation C should also be similar between the sites in the absence of nutrient limitation. It was better reproduced using **dbLang** than **siLang**, as less P limitation is simulated using **dbLang.**

Secondly, studies have shown that biological processes dominate the C-P interactions in P-poor beech forests (Bünemann et al., 2016; Pistocchi et al., 2018), which was only supported by the sensitivity results of **dbLang** (Fig. 5 and Table S3). While in **siLang**, slow sorption/desorption process dominated GPP and vegetation C regardless of P availability. Most importantly, **siLang** overestimated $P_{lab}$ and $P_{sorb}$ in topsoil but still greatly underestimated the plant productivity and biomass at low-P sites (COM and LUE). Collectively, these findings suggest that a double Langmuir model of Pi sorption-desorption better described

forest C and nutrient dynamics at these sites than the conventional single Langmuir model.

Yang et al. (2014)'s study using the **siLang** scheme has also shown that a doubled rate of P transfer from $P_{sorb}$ to $P_{lab}$, leads to similar increases in plant productivity and biomass as a doubled P mineralization rate. This is in line with our sensitivity



analysis of **siLang** showing that slow sorption/desorption of P has similar impacts as SOM turnover on GPP and plant biomass (Fig. 5 and Table S3), which seems also highly unrealistic at Amazon sites with depleted soil Pi.

From the perspective of Pi cycling, the parameterization of linear P exchange between $P_{lab}$ and $P_{sorb}$ in **siLang** lacked clear foundations and experimental evidence and hence largely differ among TBMs (Helfenstein et al. 2020). Similarly, we have shown that the exchange between $P_{sol}$ and $P_{lab}$ in most TBMs also largely vary and deviate from the reported ranges of experiments (Table 2). Most importantly, the assumption that $P_{sol}$ only exchanges with $P_{lab}$ has been heavily questioned by experimentalists (Helfenstein et al. 2020, Morel et al. 2000) and it also differes from earlier models (Barrow, 1983; Devau et

al., 2009; Van Der Zee and Gjaltema, 1992). The double-surface Langmuir isotherm much better represents the conceptual sorption-desorption scheme that is supported by experimental data (Frossard et al., 2000; Frossard and Sinaj, 1997; Fardeau, 1995), and also implicitly considers some slower Pi exchange processes that are currently ignored in TBMs, such as chelation and dissolution/precipitation.

### 4.3 Ecosystem responses to perturbations

The different Pi sorption-desorption schemes have not only led to different simulated Pi pool sizes, but also caused varying ecosystem responses to elevated $CO_2$ and P addition (Figs. 6 and 7). The main difference between **siLang** and **dbLang** was the speed and extent of exchange between the soluble/labile and sorbed Pi pools. The added P was quickly transferred to the $P_{lab}$ in both models, but the speed of Pi transfer to $P_{sorb}$ was much faster in **dbLang**. Because in **dbLang**, Pi directly transferred from $P_{sol}$ to $P_{sorb}$ while in **siLang**, the transfer to $P_{sorb}$ only occurred from $P_{lab}$ and at a much slower rate. As a consequence,

$P_{sorb}$ in **dbLang** stored more P after P addition and it also released more P under higher $CO_2$ concentrations and elevated plant P demand. The fast transfer of P from $P_{sol}$ to $P_{lab}$ in both models is supported by evidence from [33]P tracer studies showing that the radioactivity of $P_{sol}$ and $P_{lab}$ converged quickly (< 3 months) after [33]P tracer addition (Pistocchi et al. 2018). However, the speed and extent of transfer to $P_{sorb}$ cannot be easily confirmed with isotope signals since both [33]P and [32]P have very short half-lives (Frossard et al. 2011), but it is estimated to vary from days to weeks (Buehler et al., 2002). Nevertheless, the mathematical

and conceptual description of isotopic exchange kinetics tend to support the **dbLang** model, that the P transfer to $P_{sorb}$ took place at a timescale of months rather than years in acidic soils (Bünemann et al. 2016, Frossard & Sinaj 1997, Helfenstein et al. 2020). Hou et al. (2019), using a data assimilation approach, also concluded that P transfer to $P_{sorb}$ might happen at a much faster rate than the conventional TBM parameter values suggest.

Our advance in capturing soil Pi exchange has altered the responses of QUINCY to changing P availability, particularly at

low-P sites. The main difference of soil Pi responses to P additions between **siLang** (also **4pool**) and **dbLang** is that the novel **dbLang** model can store or release more P from the sorbed Pi pool with a faster rate than **siLang**. Particularly, at sites with high P stress (e.g. LUE), a further increase in P stress (e.g. $CO_2$ fertilization) led to an increased P mineralization in **dbLang** rather than an increased P desorption in **siLang** (Fig. 7), as **dbLang** can establish a much larger (implicit) microbial (fast SOM) pool in the topsoil to mineralize P (Fig. 3a). It was confirmed in the EucFACE experiment that both N and P

mineralization increased by as much as 200% under eCO2 (Hasegawa et al., 2016). Additionally, evidence from isotopic



studies has shown that biological processes, rather than geochemical processes, are dominant in both topsoil and subsoil of the P-poor LUE site (Pistocchi et al. 2018, Bünemann et al. 2016).

**4.4 Limitation and future directions**

The novel **dbLang** model much better represented soil Pi exchange and thus improved the C-P interactions in QUINCY, but

there was a caveat when the P stress was high. At the P-poor site LUE, the GPP and vegetation C in **dbLang** were almost twice as high as those in **siLang** (Figs. S2a and S2c). However, such a huge release of P stress by **dbLang** was still not enough for LUE to reach similar plant productivity and biomass of BBR or VES, as indicated by field evidence (Lang et al. 2017). In ecosystems with low soil P or high P stress, $CO_2$ fertilization enhances the organic P cycling processes, such as root exudation, phosphatase production, and microbial P mining (Ellsworth et al., 2017; Jiang et al., 2020; Lang et al., 2017; Pistocchi et al.,

2020). Such responses in SOM cycling are not yet implemented in QUINCY, leading to a much lower GPP and plant biomass at low-P site than others. Nor were these mechanisms described in other TBMs, leading to poor model performance in low-P ecosystems, e.g. Amazon forests (Fleischer et al. 2019) and Eucalyptus forests (Medlyn et al. 2016). Recent developments in soil models/modules have endeavored to improve the model description of SOM cycling by including organo-mineral association process, explicit microbial dynamics, enzyme dynamics or allocation, mycorrhizal association, etc. (Huang et al.,

2021; Yu et al., 2020a; Ahrens et al., 2020; Wutzler et al., 2017; Tang and Riley, 2014; Sulman et al., 2014; Wang et al., 2013). We believe that coupling novel **dbLang** scheme with a more mechanistic representation of SOM cycling will open the door to model the different forest ecosystem P cycling strategies (Lang et al. 2017) as well as to simulate the responses of low P ecosystems to elevated $CO_2$ (Jiang et al., 2020).

**5 Conclusions**

In this study, we first reviewed the implementation of the soil Pi exchange process in terrestrial biosphere models and compared the model implementations of P sorption with batch sorption experiment data. We found that the parameterizations used by most TBMs strongly underestimate the soil's P sorption capacity and overestimate the half-saturation Langmuir coefficient compared to the experimental data. In the QUINCY model, such a formulation leads to a much lower soil P binding capacity in TBMs than in reality, causing unrealistically large constraints of slow Pi exchange process on GPP and plant biomass,

regardless of actual soil P availability. We presented a novel model scheme, based on a double-surface Langmuir isotherm, to describe the soil Pi exchange in better accordance with data from sorption experiments. This model parameterization better simulated the measured soil Pi pools and the GPP and plant C patterns of our study sites. It also better reproduced the topsoil SOM pools in the low-P site, therefore better simulating the responses of P pools/fluxes to elevated $CO_2$ than conventional TBMs. The novel double-surface Langmuir approach can thus serve as a better modelling tool to understand ecosystem

response to global change.



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

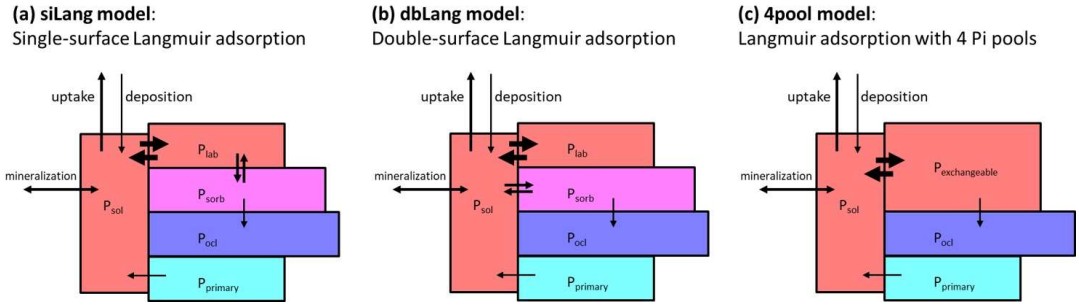

**Figure 1: Model concept of (a) conventional Langmuir model (siLang), where only the labile P pool ($P_{lab}$) is exchanging P with the soil solution ($P_{sol}$) (b) double-surface Langmuir model (dbLang), where both $P_{lab}$ and $P_{sorb}$ are exchanging P with the soil solution but with different Langmuir parameters, and (c) a Langmuir model with only four inorganic P pools (4pool), where $P_{lab}$ and $P_{sorb}$**
**are combined to one pool, $P_{exchangeable}$, to which $P_{sol}$ can get adsorbed via Langmuir sorption.**

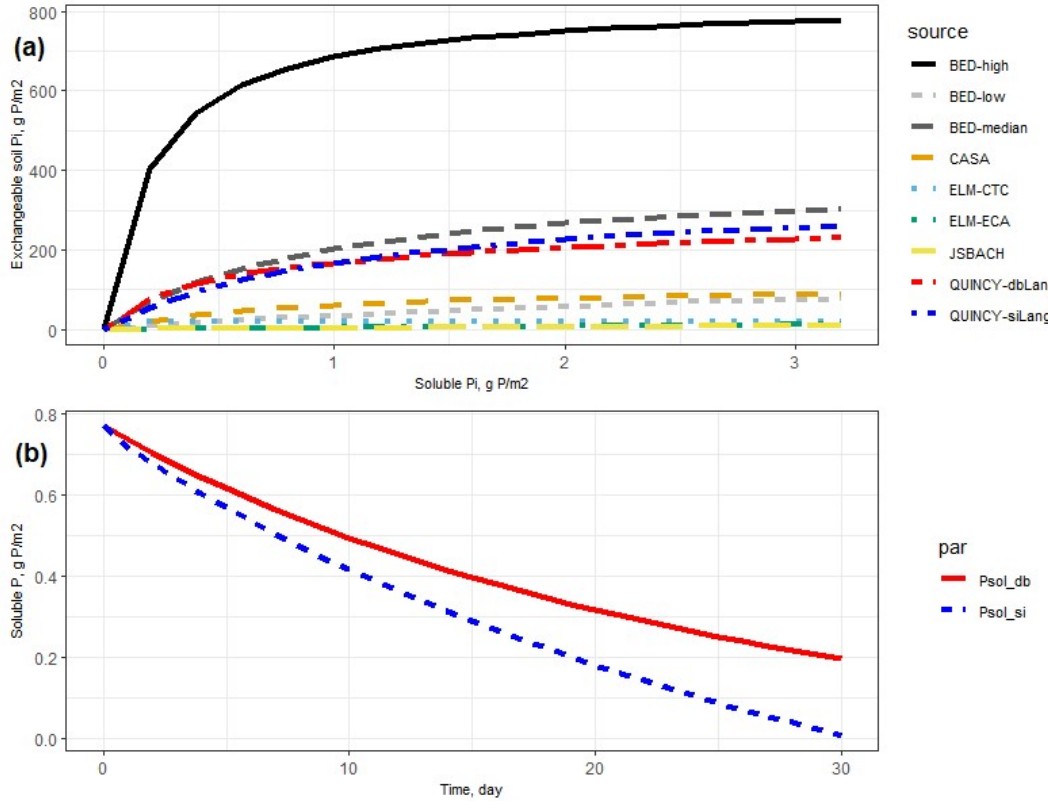





**Figure 2: (a) Exchangeable soil inorganic P ($P_{lab}$ plus $P_{sorb}$) curves based on different terrestrial biosphere models (TBMs) parameters and batch experiments data (BED). In TBMs, exchangeable soil Pi is defined as the sum of $P_{lab}$ and $P_{sorb}$. In batch experiments data, soil exchangeable Pi only refers to $P_{lab}$ in Eq. 1 due to a destructive experiment procedure which well mobilizes**

**soil Pi (Barrow and Shaw, 1979). For conventional Langmuir model (siLang), $P_{sorb}$ is calculated as 9/8 (global average ratio between $P_{sorb}$ and $P_{lab}$, reported in Yang et al. 2013) of $P_{lab}$ at all soluble P concentrations; for double-surface Langmuir model (dbLang), $P_{sorb}$ is calculated following Eq. 3.2. (b) Simulated desorption curves for single- and double-surface Langmuir isotherms. The two desorption curves start with the same exchangeable soil Pi (152.2 g P/m², $P_{lab}+P_{sorb}$) and soluble Pi (0.769 g P/m², $P_{sol}$) contents, and the same amount of P (2.5 g P/m²) is removed from soil solution every day. Both Langmuir isotherms are assumed to be in**

**equilibrium at a daily timestep.**

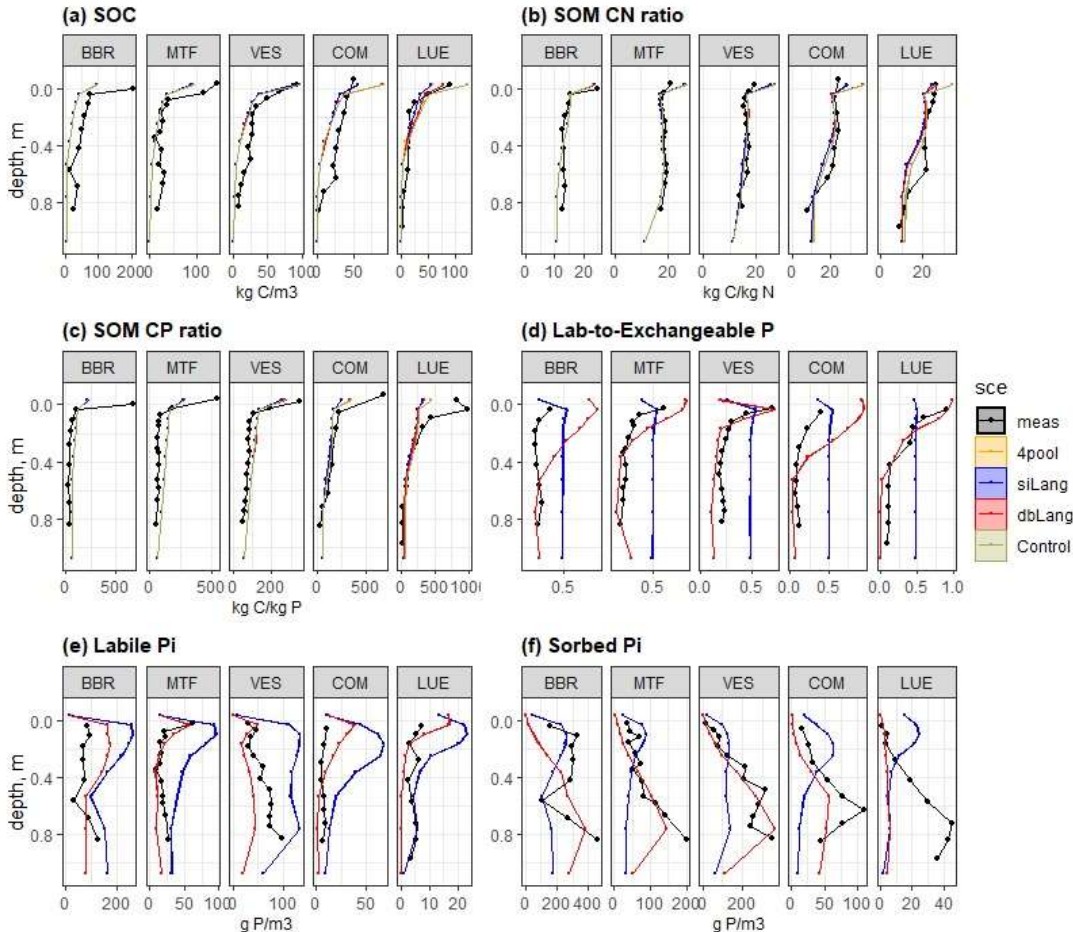

**Figure 3: Simulated and measured (a) soil organic carbon (SOC), soil organic matter (SOM) (b) CN and (c) CP ratios, (d) ratio of labile to exchangeable Pi, (e) labile Pi, and (f) sorbed Pi of the study sites along the soil P availability gradient, BBR>MTF>VES>COM>LUE. The black dotted line is the field measurements, the orange line represents the Langmuir model with**



**only four inorganic P pools (4pool), the blue line represents the single-surface Langmuir model (siLang), the red line represents the double-surface Langmuir model (dbLang), and the dark yellow line represents the Control model. 4pool and Control are not applicable in (d), (e), and (f).**

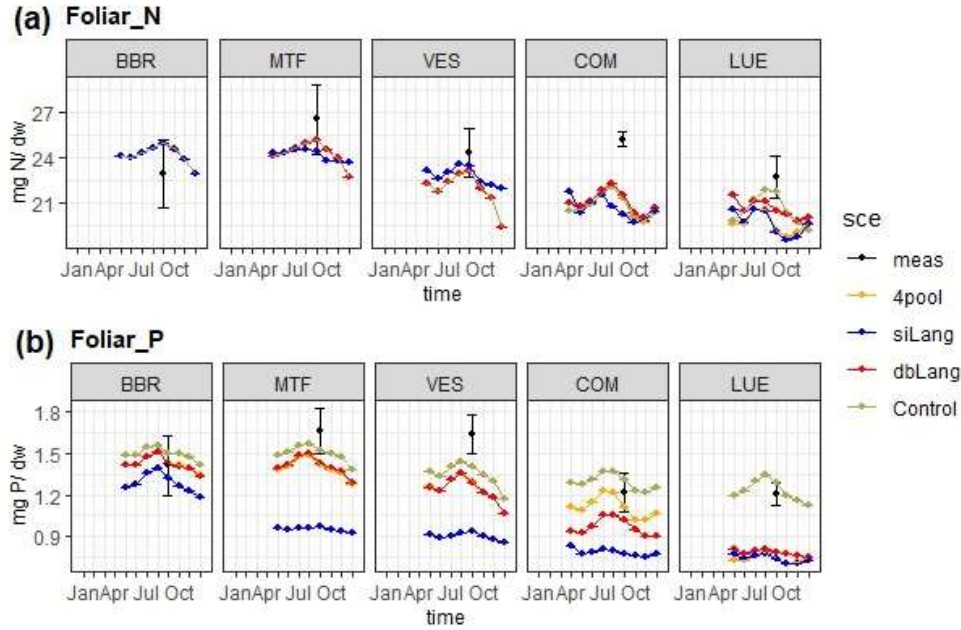

**Figure 4: Simulated and measured foliar N (a) and P (b) contents along the soil P availability gradient. The simulated values are the**
**yearly average of the period 2006–2015, and the measured values are sampled in 2014. The black dots are the field measurements, the orange line represents the Langmuir model with only four inorganic P pools (4pool), the blue line represents the single-surface Langmuir model (siLang), the red line represents the double-surface Langmuir model (dbLang), and the dark yellow line represents the Control model.**



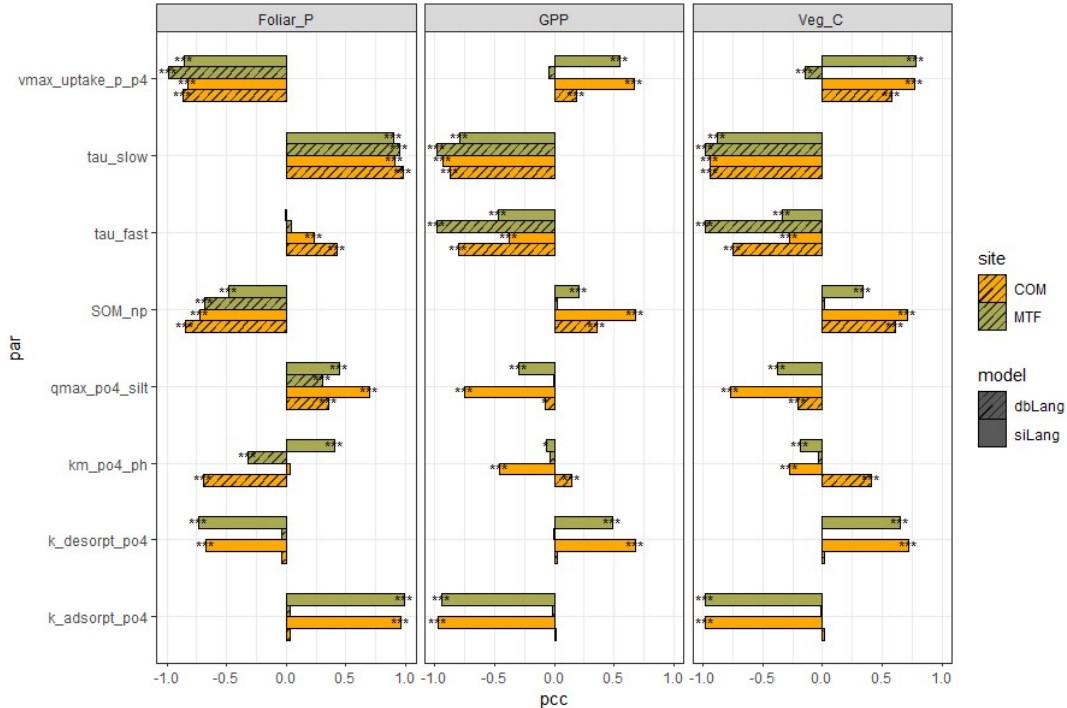

Figure 5: Partial correlation coefficient (pcc) values of eight main parameters on foliar P content, gross primary production (GPP) and vegetation C stock at COM and MTF sites of the single-surface Langmuir model (siLang) and the double-surface Langmuir model (dbLang) in the Latin hypercube design (LHS) sensitivity runs. The eight parameters are: maximum plant P uptake rate (vmax_uptake_p_p4), turnover rates of slow and fast soil organic matter (SOM) pools (tau_slow and tau_fast), N: P ratio of the slow SOM pool (SOM_NP), phosphate sorption capacity of fine soil (qmax_po4_silt), correction coefficient of pH on Langmuir $K_m$ (km_po4_ph), and the absorption and desorption rate between $P_{lab}$ and $P_{sorb}$ (k_adsorpt_po4 and k_desorpt_po4). Significance level in figure: p<0.001 (***), p<0.01 (**), and p<0.05 (*).





**Figure 6: Simulated changes of gross primary production (GPP), aboveground C, fine root C, leaf area index (LAI), uptake of N and P, foliar N and P contents to CO₂, P, and CP fertilizations at (a) BBR, (b) VES, and (c) LUE sites and (d) the ratio of unfertilized scenarios of each model to the Control model. Bars are calculated as the changes in percentage in subplots (a) to (c), and bars in subplot (d) are fractions representing how much the models deviate from a non-P-limited condition.**





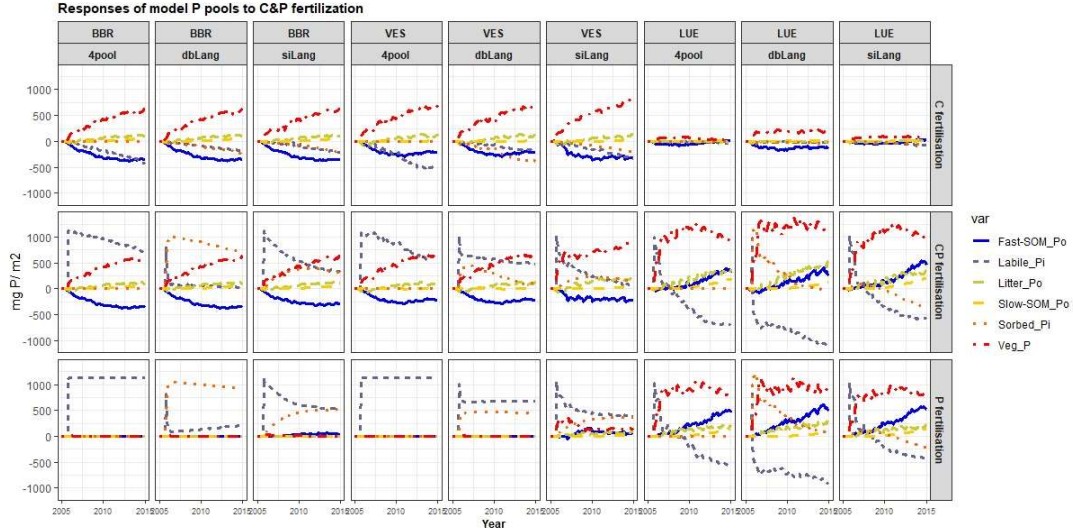

**Figure 7: Simulated temporal responses of model P pools (labile Pi, sorbed Pi, litter Po, Po in fast and slow soil organic matter (SOM)**
**pools [fast-SOM_Po and slow-SOM_Po], and P in the plant [Veg_P]) to CO₂, P, and CP fertilizations at BBR, VES, and LUE sites.**
**In the C fertilization experiment, the atmospheric CO₂ concentration was increased by 200 ppm from 2006 to 2015. In the P**
**fertilization experiment, 1139.7 mg P/m², was added once to the soil as soluble phosphate on Sep 16, 2006. The CP fertilization is the**
**combination of C and P fertilizations.**





| Study sites | BBR | MTF | VES | COM | LUE |
|---|---|---|---|---|---|
| Altitude (m a.s.l.) | 809 | 1023 | 810 | 840 | 115 |
| Mean annual temperature (°C) | 5.8 | 4.5 | 5.5 | 6.8 | 8.0 |
| Mean annual precipitation (mm) | 1031 | 1299 | 1200 | 1749 | 779 |
| Composition *beech* (*Fagus sylvatica* %) | *99* | *96* | *100* | 69 | *91* |
| Age *beeches* (a) | 137 | 131 | 123 | 132 | 132 |
| Height *beech* (mean basal area tree) (m) | 26.8 | 20.8 | 29.3 | 27.6 | 27.3 |
| Diameter at breast height *beeches*(cm) | 36.8 | 37.6 | 40.1 | 39.9 | 27.5 |
| Standing volume (m³ ha⁻¹) | 495 | 274 | 550 | 685 | 529 |
| Soil measurements in topsoil (0−30 cm) / subsoil (30−100 cm) | | | | | |
| Texture (topsoil) (WRB 2015) | Silty clay loam | Loam | Loam | Loam | Loamy sand |
| Texture (subsoil) (WRB 2015) | Loam | Sandy loam | Sandy loam | Sandy loam | Sand |
| Soil organic carbon (kg m⁻²) | 18.7/17.6 | 14.2/13.1 | 12.9/8.2 | 13.1/8.8 | 8.9/5.4 |
| Soil C:N | 14.1/13.2 | 18.2/18.3 | 16.3/16.2 | 22.7/18.2 | 23.6/16.7 |
| Soil N:P | 3.6/2.4 | 5.1/3.3 | 6.2/4.4 | 8.5/5.7 | 16.5/3.7 |
| $P_{lab}$ (g m⁻²) | 24.6/51.0 | 8.3/13.2 | 15.3/46.2 | 2.7/4.0 | 2.0/2.5 |
| $P_{sorb}$ (g m⁻²) | 94.5/181.1 | 19.4/82.3 | 39.2/167.4 | 9.2/40.5 | 2.0/19.9 |

**Table 1: Site characteristics of the study sites Bad Brückenau (BBR), Mitterfels (MTF), Vessertal (VES), Conventwald (COM), Lüss (LUE). Reproduced from Lang et al. (2017).**

| Model/Source | $S_{max}$ (g/m²) | $K_m$ (g/m²) | $k_p$ | Reference |
|---|---|---|---|---|
| CASA-CNP | 10 − 112 | 0.45 − 1.35 | 0.004 − 0.04 | Wang et al. 2007 |
| CABLE | 91 ± 42 | 55 ± 23 | 0.40 ± 0.15 | Wang et al. 2010 |





| JSBACH | $91 \pm 36$ | $60 \pm 20$ | $0.42 \pm 0.13$ | Goll et al. 2012 |
|---|---|---|---|---|
| ELM-CTC | 10 | $0.00035 - 0.005$ | $0.002 - 0.008$ | Yang et al. 2014 |
| ELM-ECA | 133 | 64 | 0.32 | Zhu et al. 2016 |
| ELMv1 | $91 \pm 42$ | $55 \pm 23$ | $0.40 \pm 0.15$ | Zhu et al. 2019 |
| ORCHIDEE-NP | NA | | $0.2 - 0.4$ | Goll et al. 2017 |
| QUINCY | $90 - 650$ | $0.15 - 2$ | $0.004 - 0.03$ | Thum et al. 2019 |
| BED -mean | $701 \pm 11$ | $6.7 \pm 0.4$ | $0.023 \pm 7e\text{-}4$ | SI Ref. [1 – 27] |
| BED -25 quartile | $187 \pm 7$ | $0.21 \pm 0.14$ | $0.005 \pm 2e\text{-}4$ | SI Ref. [1 – 27] |
| BED -median | $390 \pm 15$ | $0.93 \pm 0.06$ | $0.01 \pm 3e\text{-}4$ | SI Ref. [1 – 27] |
| BED -75 quartile | $829 \pm 41$ | $4.5 \pm 0.4$ | $0.022 \pm 0.0015$ | SI Ref. [1 – 27] |

**Table 2: Maximum sorption capacity ($S_{max}$) and Langmuir coefficient ($K_m$) as well as the calculated partition coefficient ($k_p$) of phosphorus Langmuir (ad)sorption isotherm in TBMs and batch sorption experiments data (BED).**

| Soil profiles | $K_{nrmsr}$ diff | p value |
|---|---|---|
| SOC | $0.009 \pm 0.023$ | 0.245 |
| SOM CN ratio | $0.005 \pm 0.005$ | 0.059 |
| SOM CP ratio | $0.010 \pm 0.014$ | 0.103 |
| Bulk density | $0.013 \pm 0.020$ | 0.129 |
| SOP | $-0.016 \pm 0.018$ | 0.919 |
| SIP | $0.058 \pm 0.051$ | **0.044*** |
| Labile Pi | $0.030 \pm 0.149$ | 0.356 |
| Sorbed Pi | $0.096 \pm 0.137$ | 0.118 |
| Lab-to-Exchangeable P | $0.130 \pm 0.077$ | **0.014*** |



**Table 3: Results for paired t-test between the normalized root mean square ratios ($K_{nrmsr}$, Table S1) of siLang and dbLang models of measured soil profile properties at all study sites. The null hypothesis is that dbLang performs no differently than siLang (i.e.**
**dbLang has the same $K_{nrmsr}$ values as siLang), and the hypothesis is rejected when the p value is smaller than 0.05 (values in bold with an asterisk).**

**Acknowledgments**

This work was supported by the Swedish government-funded Strategic Research Area Biodiversity and Ecosystems in a Changing Climate, BECC. SC was supported by the European Research Council (ERC) under the European Union's Horizon

2020 research and innovation programme (QUINCY; grant no. 647204). We are grateful to Prof. Dr. Friederike Lang, Jaane Krüger, and other co-workers from the German Research Foundation (DFG) funded priority research programme SPP1685 "Ecosystem Nutrition" for measuring and sharing the data, and to Dr. Jan Engel for technical assistance in developing the code.

**Author contributions**

LY designed the study. LY and SZ formulated the paper outline. LY performed the literature review, model development, and analyses. SC, LY, and SZ developed the QUINCY model that forms the basis of the analysis in this manuscript. All authors contributed to writing the manuscript.

**Data availability**

The data in the literature review is available under request to the corresponding author.

**Code availability**

The scientific part of the code is available under a GPL v3 licence. The scientific code of QUINCY relies on software infrastructure from the MPI-ESM environment, which is subject to the MPI-M-Software-License-Agreement in its most recent form (http://www.mpimet.mpg.de/en/science/models/license). The source code is available online (https://git.bgc-jena.mpg.de/quincy/quincy-model-releases), but its access is restricted to registered users. Readers interested in running the
model should request a username and password from the corresponding authors or via the git-repository. Model users are strongly encouraged to follow the fair-use policy stated on https://www.bgc-jena.mpg.de/bgi/index.php/Projects/QUINCYModel.