# Peer review of "Improved representation of phosphorus exchange on soil mineral surfaces reduces estimates of phosphorus limitation in temperate forest ecosystems"

_Biogeosciences, 2022_

## Author Response (AR1)

**Reply to "Comment on bg-2022-114 RC1"**

**RC1:** In this study, Lin et al presented a double surface Langmuir adsorption isotherm in the QUINCY model and compared it to the traditional/simplified (single surface) Langmuir isotherm, that is mostly used in current TBMs. After model calibration, the authors argue the double isotherm shows a better representation of the inorganic P cycling. The improved P (ad)sorption model also suggests the current assumed P limitation in temperature forests was likely overestimated. Overall, the paper is well written, and the results are also sound. Given our current understanding of P limitation is still very limited, thus before the P models can be applied to make sound predictions, the model structure needs to be evaluated and discussed. Study like this paper thus contributes to improving the process description of P exchange in forest ecosystems and advances in C-P coupling in TBMs. I would thus recommend publishing this work at Biogeosciences. However, I have the following comments for the authors to consider during their revisions.

First, the authors did a literature survey to highlight uncertainties of the current parameterization of Langmuir isotherm in some TBMs. The parameter error in some previous TBMs was also noticed by myself when I develop a recent ecosystem CNP model.

**Authors**: we thank the referee for acknowledging our work.

**RC1:** Thus I think it is important to highlight this for the community and a very good motivation for the current study. However, one would ask if this is just a parameterization issue or if it is a model structure issue (as the authors argued here)? Empirical data that fit different isotherm functions, including traditional Langmuir isotherm, generally show various but reasonably well-fit results (e.g. Brenner et al 2019, Lin et al. 2020, and much more).

Thus, I would like to discuss this with the authors. First, which part of the improved model fits the measured data that could be attributed to the model structure, which part is from improved parameters? I have concerns about how the model comparison is made and how much conclusions can be drawn from such a comparison? In Line 138, the authors state separate calibrations were made for each site and each depth, what's the influence of those separate calibrations for the comparison?

**Authors**: we would argue the different simulation results, especially the simulated soil inorganic P pools (Fig. 3), are solely caused by the model structure rather than parameter values. We did not quantify the effects of model structure versus parameterizations, but in one of the sensitivities test we did, it is very clear that the simulated pattern of labile P to exchangeable P ratios (Fig. S7) is rather consistent within the same model structure regardless of the parameterizations. As for the calibration mentioned in Line 138, we only calibrated against the SOM stoichiometry to minimize the impact from organic cycling processes and did not calibrate the double Langmuir parameters (which are based on soil texture, pH, and exchangeable Al/Fe of each study site), thus we believe the different results we showed are caused by model structure rather than parameterizations.

**Revision: removed line 137. At line 139 adding: '**All the other parameter values were either taken from Thum et al.'s (2019) study or modified based on the development in this study, except for the maximum biological N fixation rate, which is set to be 2.5 kg N ha$^{-1}$ yr$^{-1}$ for all the study sites.'

**RC1:** Second, as the authors argue the advantage of using double surface Langmuir, i.e. its higher buffer capacity. Then I would suggest a better separation of the influence of P release from other releases and uptakes? i.e. the feedback is of need. From the results, the main improvement is the ratio between Plab and exchangeable Pi (section 3.2). The P uptake across models seems rather similar (Fig. S2), i.e. for the P limited site LUE, the uptake PO4 for siLang, dbLang and 4pool model (Fig. 2Sf). The siLang shows a higher uptake in autumn but at an annual scale, the overall rates seem rather similar. The different approaches show a rather large influence on the C partitioning, (LAI, aboveground C, Fig. S2 bc). This is rather strange, what causes such large feedback on the aboveground plant properties, given the total P uptake seems rather similar? I also do not find evidence to support the statements of Line 203, i.e. the differing plant P uptake.

**Authors**: the plant P uptake in QUINCY is controlled by multiple factors, including soluble P concentration, root biomass, and plant P demand. Moreover, the root biomass is also heavily influenced by productivity and the plant nutrition status. We totally agree that it is not straight forward to directly draw any conclusions from the plant P uptake comparison, as the referee pointed out in Fig. S2, the rather similar annual P uptake could be either due to a combination of high root biomass and low soluble P (siLang, VES), or low root biomass and high soluble P (control, LUE). In general, high root biomass lead to higher uptake in non-growing season but meanwhile probably lead to lower uptake in growing season as the high root biomass indicates strong nutrient deficiency, i.e. very low soluble P concentration in growing season. We will clarify it more in the revision.

As for Line 203, we should refer to Fig. 4 rather than Figs. S2-5. We thank the referee for pointing this out and will change it in the revision.

**Revision:** from line 203 to 210: "The simulated difference in foliar P content across models reflects both plant P uptake and productivity (Figs. 4 and S2–5). For example, at the P-rich BBR site, although the simulated gross primary productivity (GPP), leaf area index (LAI), aboveground C, and fine root C for the four models were almost identical (Fig. S2), the foliar P contents of four models were different due to differing plant P uptake (0.99, 0.95, 0.89, 0.95 g P/m$^2$/yr. for **Control**, **dbLang**, **siLang**, and **4pool**, respectively; Fig. S2). In contrast, at the P-poor LUE site, the differences among **siLang**, **dbLang** and **4pool** in GPP, LAI, and plant C were more pronounced than that in foliar P content, because the limited P uptake (0.67, 0.53, 0.55 g P/m$^2$/yr. for **Control**, **dbLang**, and **siLang**, respectively; Fig. S2) drastically changed the plant C allocation (e.g. leaf to root ratio) and led to the subsequent changes in plant properties (Figs. 4 and S2)."

**RC1:** Third, the model performance of foliar P, Fig.4b shows a convergence of different models when P availability becomes smaller. In other words, in more P-limited conditions, the difference between the models becomes smaller, although all of them largely underestimated the measured P concentration. How come such large differences in the Prich sites? Is this due to the calibration being mainly focused on the soil and thus less on the vegetation?

**Authors**: The main reason for the small difference of foliar P at low P sites by different model variants, as we have already touched upon a bit in the discussion (Sect. 4.4), is that the main mechanism to alleviate the simulated P stress in the low P site should be the organic P cycling rather than the inorganic P cycling. With this said, the double Langmuir isotherm we presented

here would help to release some of the unrealistic P stress in conventional TBM, but without an advanced organic P cycling scheme implemented, it is still not possible to reproduce the observed pattern in foliar P along this P gradient.

**RC1:** Some more specifics to consider:
Introduction
Line 36, missing references after "boreal forests are generally considered N limited"
**Authors**: thanks, we will include it in the revision
Revision: line 36 "(Lebauer and Treseder, 2008)" added
Line 49-50, the argument is that organic P recycling is the major flux, while the geochemical P flux is small.
**Authors**: thanks, we will revise it
**Revision**: line 49-50 : "Many modelling studies emphasize the significance of biological P processes (Fleischer et al., 2019; Jonard et al., 2010; Yu et al., 2018; Wang et al., 2010) and underestimated the role of geochemical P processes. In these models, organic P recycling processes are the major fluxes, while the geochemical P fluxes are small (Sun et al., 2020). Particularly, the effect of (ad)sorption kinetics was seldom discussed in previous modelling studies (Fleischer et al., 2019; Yang et al., 2014), although they are known to directly and strongly regulate $P_{sol}$ and $P_{lab}$, and thus affect P bioavailability (Frossard et al., 2000; Shen et al., 2011)."

Line 50-55. In literature, several isotherms, or model functions, including double Langmuir, have been suggested to describe the phosphorus adsorption-desorption processes (i.e., McGechan and Lewis, 2002). I would also suggest not to use "a novel model concept, Line 54" as the authors propose in its current form. It has been in the literature for some time. I think the novelty is the implementation of the TBM models and evaluation of the implications? Besides, I am also lacking the field and experimental evidence to support the additional supplement of P from the adsorbed P pools. So, what is the, i.e. P isotopic data suggest, and do they support your hypothesis here? What are the mechanisms behind that? I would suggest adding those to the motivate current model development work.
**Authors**: thanks. Yes, the isotherm itself is not novel but the implementation in TBM is. We will clarify it in the revision. The main experimental evidence to support this isotherm is the P isotopic studies, and our scheme is also based on their conceptual model (Line 266-270).
**Revision:** line 54: removed "and applied" and "novel"
Line 58-59: "We hypothesised that both Plab and Psorb exchange with Psol in the new model (Fig. 1b), following the recent evidence from isotopic studies (Helfenstein et al., 2020; Frossard et al., 2011)."

**RC1:** Methods
Line 70, equ 1, the Langmuir isotherm, do the interaction with water considered? As the concentration also dependent on the water content at each time step?
**Authors**: yes, the concentration is water dependent.
Line 116 do you have leaf P/N concentration data over years? Or just sampled for one year?
**Authors**: we have only multiple year data for one site, not for all the sites
Line 138 what is calibrated and what criteria were used for the calibration? Be specific here.

**Authors**: See replies above. we will clarify it in the revision.

**Revision**: line 139 adding "All the other parameter values were either taken from Thum et al.'s (2019) study or modified based on the development in this study, except for the maximum biological N fixation rate, which is set to be 2.5 kg N ha$^{-1}$ yr$^{-1}$ for all the study sites."

**RC1**: Results

Line 203 given the total P uptake by different approaches?

**Authors**: yes

**Revision**: from line 205: "For example, at the P-rich BBR site, although the simulated gross primary productivity (GPP), leaf area index (LAI), aboveground C, and fine root C for the four models were almost identical (Fig. S2), the foliar P contents of four models were different due to differing plant P uptake (0.99, 0.95, 0.89, 0.95 g P/m$^2$/yr. for **Control**, **dbLang**, **siLang**, and **4pool**, respectively; Fig. S2), . In contrast, at the P-poor LUE site, the differences among **siLang**, **dbLang** and **4pool** in GPP, LAI, and plant C were more pronounced than that in foliar P content, because of the effect of limited P uptake (0.67, 0.53, 0.55 g P/m$^2$/yr. for **Control**, **dbLang**, and **siLang**, respectively; Fig. S2) drastically changed the plant C allocation (e.g. leaf to root ratio) and led to the subsequent changes in plant properties (Figs. 4 and S2)."

Line 246 the pool sizes differs also after the simulation, SOM top soil, the fluxes and the pools sizes. As also your sensitivity results show the importance of SOM pools for dbLang, Line 219, which indicates the potential feedback due to the biological mineralization. Also in your Line 243 on the plant and soil changes

So the different approaches show impacts on the fluxes and pool sizes. Why not show a complete P budget for each site with different fluxes simulated by various approaches? Also show the different pool sizes before the simulation and after the simulation, i.e. the pool size changes. This will give an overall picture of the ecosystems.

**Authors**: yes, we do see an effect on SOM size and fluxes in topsoil at LUE site, but the difference for other sites are not so strong. Fig. 7 shows the temporal change (10 years) of main P pools of different models after different fertilization experiments, we believe it is a more intuitive display than a P budget for a certain period. In this figure, each point in the x axis represents a P budget at a specific time, with the size change of all P pools equal to zero (C fertilization) or the added P (P and CP fertilization). The change of the P pool also infers the changes in P fluxes, e.g. plant P pool change indicates plant P uptake change, labile P pool change indicates P adsorption change and etc.

**Revision**: caption of figure 7 adding: "Each point in the x axis represents a P budget change at the specific time, with the size change of all P pools equal to zero (C fertilization) or the added P (1139.7 mg P/m$^2$, P and CP fertilization). The change of a specific P pool at any given time equal to the total change of related P fluxes since the fertilization, e.g. plant P pool change relates to plant P uptake and P litterfall, labile Pi pool change relates to P adsorption (and P absorption) (Fig. 1), Po in fast SOM pool change relates to litter and fast SOM decomposition, and etc."

Some references mentioned:
McGechan and Lewis, 2002 presented an excellent review of the principles, equations, and models for the sorption of phosphorus. Biosystems Engineering 82 (1), 1-24.
Brenner, Julia, et al. 2019, Phosphorus sorption on tropical soils with relevance to Earth system model needs, Soil Research, 57, 17-27.
Lin Yang, et al 2020, Anoxic conditions maintained high phosphorus sorption in humid tropical forest soils, Biogeosciences, 17, 89-101.

**Authors**: thanks, we will include them in the revision

**Reply to "Comment on bg-2022-114 RC2"**

This paper by Yu et al. describes a new algorithm to better represent soil phosphorus sorption dynamics in a terrestrial biosphere model – QUINCY. The authors proposed the use of a double-surface Langmuir isotherm to better capture the non-linear relationships between solution P and labile P pools in the soil. They performed a review on both published data and model assumptions on P sorption. They then compared their simulation against data at a range of P availability sites, performed sensitivity on parameters and compared simulations for CO2 and P enrichment scenarios. They argued that the double-surface Langmuir isotherm is a better modeling scheme because it simulated observed pattern of soil organic pools well, it maintained a relatively stable solution P pool to act as a buffer against instability, which then led to less P limitation at the P-poor site, and it led to improved simulation of folia N and P concentration.

**Authors**: we thank the referee for acknowledging our work.

Overall, this is a clearly-written manuscript. The rationale and objectives are crystally clear. The discussion is also well written. My comments mostly focus on two aspects of the results that I want to discuss with the authors and receive their clarifications:

- Dd it indeed lead to improved estimate relative to the conventional single surface approach? All models performed well for reproducing the measured SOC etc. as reported in figure 3. The novelty of the double-surface scheme, as the authors argued, is that it better reproduced the ratio between Plab and exchangeable Pi (L190-191; Table 3). But looking at Table 3, the statistical significance is relatively weak ( p = 0.014 for lab-to-exchangeable P ratio, and 0.044 for SIP). At the same time, I wonder if the new scheme actually increase model complexity or not. May be the authors should make a paragraph discussing whether the gained benefits in terms of improved simulation accuracy is worth the added complexity, if there's indeed additional complexity associated with the new scheme. In particular, does it require additional parameters relative to the conventional approach? And, if we want to constrain the parameters in the new model scheme, what data collection should we make? If it doesn't involve additional complexity, I think it's very useful to highlight.

**Authors**: As we specifically calibrated the SOM profile for each site at each depth to avoid the side-effects of organic cycling, the good model fit of SOM profile in Fig.3 is the outcome of model calibration. The main improvement in Fig. 3 and Table 3, is the improved soil inorganic P simulation.

The new double Langmuir isotherm in QUINCY does require more parameters compared to other TBMs, but in this study, the siLang and dbLang shared the same set of parameters of QUINCY, i.e. the siLang also takes account the effect of soil texture, pH and extractable Al/Fe in the P sorption parameterization. Therefore, the dbLang complexity is the same as siLang in this paper, but unfortunately, we did not compare the QUINCY parametrizations with other TBMs in this study, as shown in Table 2, the original version of QUINCY (Thum et al. 2019) used Qmax and Km values which are very different from other TBMs. It is mostly because we already use soil texture and SOM content to parameterize the Qmax and Km in the original QUINCY, as many other TBMs directly use prescribed values.

The new parameterizations are dependent on soil texture, pH, and extractable Al/Fe. As far as we know, the biggest difficulty for upscaling is the extractable Al/Fe.

**Revision**: lin293 adding "Additionally, the different performance of **siLang** and **dbLang** in this study shared the same set of parameters of QUINCY thus had the same model complexity. However, the new scheme did require more input than other TBMs in Table 2 as the new parameterizations were dependent on soil texture, pH, and extractable Al/Fe, while most other TBMs use prescribed values."

- What does it mean for the land C sink estimates under future rising $CO_2$ if the model simulated a less P limitation at the P-limited site. As the authors introduced, there has been a lot of model development to add P-cycle into models. The relative magnitude of the P limitation is obviously different, but one of the crucial argument for the inclusion of P-cycle in models is that they would impose additional processes to constrain ecosystem productivity for P-poor regions of the world. The new scheme seemed to alleviate the extent of P limitation, and therefore I wonder how does it compare to a simulation without the P-cycle turned on. Do you obtain similar $CO_2$ responses for the Plimited site? Obviously the CN-only simulation does not have the capacity to accurate reflect the processes limiting $CO_2$ responses at the P-poor site, but it would be interesting to see if there's indeed difference between the two approaches.

**Authors**: thanks for the very intriguing point. The initial motivation for this study is that, the field observations of our five study sites (along a soil P gradient) do not show a clear trend in biomass, productivity, or foliar P content, as most of the observed differences were seen in soils, such as the SOM stocks and quality, the root biomass, the microbial biomass, and the root/litter P contents. It indicates that the temperate ecosystem has mechanisms to adapt to the varying soil P availability, and such mechanisms were not represented in our TBMs yet. In other words, the simulated high P stress by siLang model does not exists in reality.

The double Langmuir isotherm alleviate the P stress, but only to a certain extent. Because it only releases the buffering capacity of soil mineral surfaces, but not yet changes the organic P cycling part, which is believed to be the dominant processes in the low P system. We specifically discuss it in Sect. 4.4 to point out that this new scheme is not enough to resolve the P cycling features in extremely low P ecosystems, such as Amazon or Australian eucalyptus.

We believe the main processes regulating the P limitation are not yet implemented in the TBMs yet, such as the regulation of microbial cycling, the phosphatase production, the resorption within plant and etc. We have been working towards this direction, but as the referee mentioned, the control simulation in our study only reflects the $CO_2$ responses induced by the current model processes, and it is a promising future direction to include and quantify the effects of these processes.

Lastly, one question I have, which isn't a criticism per se, is that why the authors didn't use the Jena Soil Model (that they developed) to investigate the effects of different soil P sorption functions. It appears to be a soil question and therefore I wonder is there any particular reason that deemed JSM unsuitable for this work?

**Authors**: at the time when this study started, the JSM is not yet fully functional with QUINCY vegetation processes. Now it is possible to test different sorption functions with JSM enabled, and it is for sure something we would like to test in future that what is the role of inorganic verse organic cycling processes given different soil P availabilities, as we believe more mechanisms of organic P cycling processes are included in JSM which would facilitates such tests.

Specific comments:

Title: replace P with phosphorus.

**Authors**: thanks, will revise

Figures: Figure presentation needs to improve quite significantly. Figures are currently in low quality resolution. Units and variable names in the figure legend/axis need to be properly labeled.

For Figure 3, the authors may need to think of a better way to show the results, as it's not very clear to see differences in panel a and b (but maybe because they are similar and therefore it's not important to show the differences?). But still, the 4pool and control color are too hard to differentiate from each other.

**Authors**: thanks, we will remake the figures to better present the results.

**Revision**: we have replotted Figs. 3, 4, 7, and S2-5, S7 with more distinctive color schemes and higher resolutions. Figure 3 was also rescheduled into two panels, one showing the SOM profiles with four models, and the other showing the soil inorganic profiles with only **siLang** and **dbLang** models.

Table 3. One could argue the statistical significance is very weak.

**Authors**: yes, agreed. since we only have 5 sites, it is not very likely to yield a very low p values in the paired-t test. Fig. 3 presents the visual difference between different model performances, but we think it might still be necessary to conduct a simple statistic analysis to confirm the pattern. We will be careful in interpreting the results.

**Revision**: line 289 adding: "Such an improvement of **dbLang** in reproducing the *Lab-to-Exchangeable P* ratio much better captures the vertical patterns of soil inorganic pools (Figs. 3d-f), which is also supported by statistical analysis that the measured soil P profiles were slightly better reproduced (Tables 3 and S1). This was not simply caused by improvements in modelling the individual labile or sorbed Pi pool, but rather by a better representation of the Pi exchange among $P_{sol}$, $P_{lab}$, and $P_{sorb}$, as such an improvement is independent of site conditions and model parameterization (Figs. 3 and S7)."